# Ribonucleoside Hydrolases–Structure, Functions, Physiological Role and Practical Uses

**DOI:** 10.3390/biom13091375

**Published:** 2023-09-12

**Authors:** Leonid A. Shaposhnikov, Svyatoslav S. Savin, Vladimir I. Tishkov, Anastasia A. Pometun

**Affiliations:** 1Bach Institute of Biochemistry, Federal Research Centre “Fundamentals of Biotechnology” of the Russian Academy of Sciences, Moscow 119071, Russia; savinslava@gmail.com (S.S.S.); vitishkov@gmail.com (V.I.T.); 2Department of Chemical Enzymology, Chemistry Faculty, Lomonosov Moscow State University, Moscow 119991, Russia

**Keywords:** nucleoside hydrolases, crystal structure, catalysis mechanism, enzyme kinetics, cancer drug design

## Abstract

Ribonucleoside hydrolases are enzymes that catalyze the cleavage of ribonucleosides to nitrogenous bases and ribose. These enzymes are found in many organisms: bacteria, archaea, protozoa, metazoans, yeasts, fungi and plants. Despite the simple reaction catalyzed by these enzymes, their physiological role in most organisms remains unclear. In this review, we compare the structure, kinetic parameters, physiological role, and potential applications of different types of ribonucleoside hydrolases discovered and isolated from different organisms.

## 1. Introduction

With technological progress, the sequencing of the genomes of various organisms has turned from an expensive and rare procedure into an affordable and routine one. Currently, many genomes of various organisms from all three kingdoms have been sequenced, in which key proteins and enzymes with important functions have been annotated. However, the study of the key enzymes of such a vast set of organisms is a laborious process, because it is important not only to find the enzyme but also to study its main properties in order to understand and describe its physiological role and propose a potentially useful application for humans, be it in biotechnology, diagnostics, or medicine. In addition, some enzymes go unnoticed for a long time, because their potential benefits were not always clear at first glance and the studies carried out were not systematized. This review article is aimed at systematizing knowledge on the ribonucleoside hydrolase (Rih) family of enzymes. The last review on this topic was published in 2003 by Wim Versees and Jan Steyaert in *Current Opinions in Structural Biology*, in which the authors focused on Rih structure and the catalysis mechanism. In this work, we aim to discuss Rih hydrolases more broadly, such as the physiological role of these enzymes in various organisms, the features of the primary and spatial structure of these enzymes, their main properties and mechanism of catalysis, stability, and possible areas of practical application.

## 2. Physiological Role of Rih Family Proteins

Ribonucleoside hydrolases (Rih) (EC 3.2.2.1–3.2.2.3, 3.2.2.7, 3.2.2.8, 3.2.2.13, 3.2.2.25) are a family of enzymes that are able to hydrolyze ribonucleosides at the N glycosidic bond to ribose and the corresponding nitrogenous base (Figure 1 and Figure 2). All chemical structures in this work from here onwards were drawn using the free online service ChemDraw JS [1].

For protozoan parasites (e.g., *Trypanosoma brucei brucei*, *Leishmania major*, *Crithidia fasciculata*), the role of these enzymes in metabolism is clear. These organisms cannot synthesize nitrogenous bases de novo; they must obtain them in other ways. To do this, they receive ribonucleosides from the outside, which are further converted into nitrogenous bases by phosphorylation or by hydrolysis. To implement the second process, their genome contains genes encoding certain Rih hydrolases, and thus for these organisms, such hydrolases are a key participant in metabolism [2,3,4,5,6,7,8,9,10,11,12,13,14,15,16].

For bacteria, the physiological role of these enzymes is not completely clear. Bacteria have two common metabolic pathways for the production of purines and pyrimidines: de novo synthesis and recycling of nitrogenous bases and nucleosides. It is noteworthy that neither de novo synthesis nor the recycling route involves a simple synthesis of nucleosides from nitrogenous bases and ribose. Instead, the synthesis of nucleosides from scratch occurs in several steps, and the purine or pyrimidine bases are synthesized from several precursor molecules [17] (pp. 2883–2897). During the reutilization (salvage) path, phosphorylation and phosphorylases associated with this process play an important role [17] (p. 2916). Purine and pyrimidine nucleoside phosphorylases (purine NP, EC 2.4.2.1; pyrimidine NP EC 2.4.2.2) play a key role in the salvage pathway of nitrogenous bases [18,19]. These enzymes catalyze the reaction shown below in Figure 3.

Curiously enough, these enzymes are more commonly used in different organisms as opposed to nucleoside hydrolases for purine and pyrimidine salvage pathways, the exception being protozoa, where nucleoside hydrolases play a major role in metabolism. While the reactions of these different types of enzymes (NHs and NPs) are different, the common role is the same: making a nitrogenous base from a nucleoside. Still, only NPs are more commonly used in organisms for salvage of purines and pyrimidines, although genes of NHs are present.

According to this, ribonucleoside hydrolases do not play a key role in the metabolism of bacteria, although these genes are present in their genomes, though some spore-forming bacteria, such as *Bacillus cereus* and *Bacillus anthracis*, require nucleoside hydrolases to prevent adenosine- or inosine-induced sporulation by cleaving the appropriate nucleosides [20,21,22].

For plants, nucleoside hydrolases may be needed under stressful conditions with a lack of nitrogen, when nitrogenous bases become its source, for the production of which Rih hydrolases is needed. Under normal conditions, these enzymes do not play a key role. These hydrolases have been found in various plants [23,24,25,26,27], and in some plants these enzymes are present as several isoforms. The physiological role of Rih hydrolases in plants is not limited to the cleavage of standard ribonucleosides. Decreased nucleoside hydrolase activity in *Physcomitrella patens* leads to delayed bud formation, suggesting involvement of Rih hydrolases in cytokinin metabolism [23].

In the case of other eukaryotes, nucleoside hydrolases have been found in yeast [28], fungi [29,30,31,32,33], nematodes [34,35], insects [36], and fish [37]. In addition, genes homologous to various Rih hydrolases have been found in other eukaryotes, but the enzymes encoded by these genes have not been studied. In mammals, Rih genes have not yet been found. Nucleoside hydrolase from the *Aedes aegypti* mosquito, which is a carrier of the Dengue fever virus, is injected into the body at the site of a mosquito bite, destroying circulating adenosine, thereby preventing activation of mast cells through purinergic signaling and thus acting as a local anesthetic [36]. The yeast *Saccharomyces cerevisiae* requires nucleoside hydrolase in the metabolism of pyridine nucleoside nicotinamide riboside, a precursor of NAD^+^ [38]. Nucleoside hydrolase activity in this case contributes to the activation of sirtuins, which are important for prolonging the life of yeast cells. In other eukaryotic organisms, the role of Rih hydrolases remains unclear.

Two types of Rih hydrolases should be noted. These are 1-methyladenosine nucleosidase (EC 3.2.2.13) and N-methyl nucleosidase (7-methylxanthosine nucleosidase, N-MeNase, EC 3.2.2.25). Reactions catalyzed by them are shown in Figure 4 and Figure 5, respectively. The interesting thing about them is that both of these hydrolases can only catalyze the specific reaction below and cannot cleave any other purines [39,40,41,42,43]. The 1-methyladenosine nucleosidase has only been found in starfish [39,40,44]. It catalyzes the production of 1-methyladenine, which is responsible for inducing oocyte maturation and is found in ovaries of starfish. N-MeNase has been found in tea and coffee plants [41,42,43]. It catalyzes the production of 7-methylxanthine in these plants, which then turns into theobromine and caffeine. Thus, this nucleoside hydrolase also has an important role in the metabolism of the respective organism.

From the information above, it can be concluded that for some organisms, Rih hydrolases do not play a key role in metabolism, while for others, they play one of the key roles in vital activity or under stress.

According to their substrate specificity, these hydrolases can be divided into three categories: pyrimidine-specific, purine-specific, and nonspecific (Figure 6) (excluding 1-methyladenosine nucleosidase and N-methyl nucleosidase since they catalyze highly specific reactions, although both could be considered purine-specific). Pyrimidine hydrolases (also known as CU-NH, cytidine–uridine-preferring nucleoside hydrolases) are specific to ribonucleosides containing a pyrimidine nitrogenous base (uracil, cytosine). Currently, hydrolases of this type are called RihA and RihB (alternative names: pyrimidine-specific NH, YbeK for RihA hydrolase, YeiK for RihB hydrolase), discovered in *E. coli* and first named so by researchers in [45]. There is not much difference between these two types of hydrolases. They have very similar substrate specificity, although it was noted by the researchers in [45] that while RihA hydrolase catalyzed only the cleavage of pyrimidine nucleosides, RihB also catalyzed the cleavage of purine nucleosides, albeit at an extremely low rate (much lower rate than nonspecific RihC from the same organism). An example of a reaction catalyzed by these types of hydrolases is shown in Figure 1. Hydrolases of this type are found mainly in bacteria [45,46,47,48,49]. Hydrolases of the second type are specific to ribonucleosides with purine nitrogenous bases (adenine, guanine, xanthine, hypoxanthine). These hydrolases can, in turn, be divided into two categories: specific to all purine bases (abbreviated as IAG-NH, i.e., inosine–adenosine–guanosine-preferring nucleoside hydrolase) or specific to 6-oxopurine bases (abbreviated as IG-NH, i.e., inosine–guanosine-preferring nucleoside hydrolase). An example of a reaction for this type of hydrolase is shown in Figure 2. Hydrolases of this type are found in archaea [50,51,52,53], protozoa [3,5,12,13,14], some bacteria [20], fungi [30,33], and insects [36]. Hydrolases of the third type are nonspecific purine–pyrimidine hydrolases. Alternative names also include RihC, nonspecific NH, IU-NH (inosine–uridine-preferring NH), YaaF. These hydrolases can catalyze the cleavage of both purine and pyrimidine ribonucleosides. Most preferably, these hydrolases react with uridine and inosine, but other nucleosides are also hydrolyzed at a certain rate and specificity. Examples of reactions are shown in Figure 1 and Figure 2. Hydrolases of this type are found in a wide variety of organisms: bacteria [21,34,45,54,55,56,57,58,59,60,61,62,63,64,65,66], plants [23,24,25,26,27], yeast [28,38], fungi [29,32], metazoa [34,35,37], and also some protozoa and archaea [53].

## 3. Ribonucleoside Hydrolase Structure

### 3.1. Rih Amino Acid Sequences

The wide distribution of an enzyme and the broad substrate specificity can usually be explained by differences in structure. Sometimes, the replacement of just one amino acid residue in the catalytic center can lead to a dramatic change in the properties of the enzyme. In this regard, it is important to consider the structure of different Rih hydrolases in order to explain differences in substrate specificity and such a wide variety of organisms containing genes for these hydrolases.

To assess the evolutionary relationship of various Rih hydrolases, a phylogenetic tree was constructed using the MUSCLE algorithm [67] and processed in the Geneious Prime program [68], as shown in Figure 7. Data on organisms and specific hydrolases, as well as NCBI reference sequence codes, are presented in Table 1. A total of 88 hydrolase sequences from different organisms were selected. From closely related sequences, for example, from sequences of the same hydrolase from different strains of the same bacterial species, one was selected to increase sample diversity.

On the phylogenetic tree (Figure 7) and in Table 1, hydrolases from various organisms are marked with different colors. Some hydrolases are marked with asterisks in the table, and their name does not indicate the type of hydrolase. This was due to the fact that the type of hydrolase was not determined for these enzymes when they were made public. However, from the analysis of the phylogenetic tree and further from the alignment of amino acid sequences, it can be assumed that these hydrolases should belong to nonspecific RihC hydrolases.

Analyzing this phylogenetic tree, it is evident that hydrolases form several clusters according to the organisms in which they are found. There is a large cluster of bacterial Rih, and within this cluster one can see a small cluster of RihA hydrolases and a small cluster of RihC hydrolases, in which some RihB hydrolases are present in between. From the point of view of evolutionary development, this is logical: some organisms required hydrolases that act only on pyrimidine nucleosides, and all these RihA hydrolases are similar to each other; and some organisms needed different type of nucleosides, instead developing the ability to catalyze reactions with different nucleosides. RihB hydrolases are a more interesting subject for discussion. Despite the fact that they show the greatest efficiency in the catalysis of pyrimidine nucleosides, like RihA hydrolases, which is why they are combined with RihA hydrolases into one subtype, it was noted [45] that RihB hydrolases can also catalyze the cleavage reactions of some purine nucleosides at a very low rate. This can also be seen from the tree: they are evolutionarily closer to RihC hydrolases than to RihA hydrolases, being in between the former. Regarding bacterial Rih, it is also worth noting that among them there are some that are located far from the main bacterial cluster and are evolutionarily closer to other types of organisms. The reason for this diverse development of bacterial nucleoside hydrolases remains unclear.

Small clusters of other organisms are also clearly expressed on the phylogenetic tree: a separate cluster of animal hydrolases, a cluster of archaeal hydrolases, which are evolutionarily close to some bacterial hydrolases, and two separate unrelated clusters of yeast and fungal hydrolases. Some hydrolases from protozoa turned out to be evolutionarily close to the hydrolases of metazoa, while others turned out to be closer to bacterial ones. The most interesting on the evolutionary tree are Rih hydrolases from plants. It is known that many enzymes in plants are synthesized as several isozymes differing in primary structure and often in properties. The same is observed for hydrolases from plants *Arabidopsis thaliana* and moss *Physcomitrella patens*, each having three RihC isoenzymes. It is noteworthy that the hydrolases of these plants are in different clusters and that these clusters are evolutionarily distant from each other compared to other hydrolases. Why evolution has led to such a strong difference between one type of hydrolase in one type of organism remains not fully understood. This may be due to different ancestors of these plant types, and accordingly, different parent enzymes of current RihC hydrolases. In addition, it can be seen that some hydrolases are knocked out of their main clusters and often have elongated branches on the tree. The length of the branches indicates the average number of substitutions of amino acid residues in comparison with the neighbors, that is, in general, the proximity of the primary structure of specific hydrolases. Long branches indicate that although this hydrolase is close to other hydrolases nearby (has one tree node with them), it still differs significantly from them in its amino acid composition (the branch is elongated).

Three amino acid sequence alignments were constructed for the selected sequences. Proteins were grouped according to the type of substrate specificity: pyrimidine-specific (RihA and RihB), purine-specific (IAG-NH and IG-NH), and nonspecific (RihC). All available organisms for each protein type were selected for comparison. Alignments are presented in Appendix A.

The purine-specific hydrolases IAG-NH and IG-NH have been found in protozoa, archaea, and some bacteria. Appendix A shows the alignment of the amino acid sequences of some of these hydrolases. It can be seen that at the N-terminus of these enzymes, there is a strictly conserved DXDXXXDD sequence in the region of positions 10–20, which is responsible for the binding of calcium ion and ribose. Calcium binding is also aided by Asp residues at positions 205 and 305, while conserved GN and EXN sequences at positions 190 and 200, respectively, are involved in ribosyl ring binding [4,12,69]. The binding of the calcium ion and the ribose ring is a universal property of all ribonucleoside hydrolases. Therefore, the amino acid residues responsible for these functions are highly conserved, and this applies to all Rih hydrolases. As will be seen below, everything is much more complicated with the binding of a nitrogenous base, and accordingly the determination of amino acid residues that determine substrate specificity. In [12,14], it was suggested that residues W83 and W260 are responsible for the binding of the purine ring of purine nucleosides, explaining this by π-stacking interactions. However, in [4], studying 6-oxopurine-specific hydrolase, it is noted that there are no tryptophan residues in these structurally equivalent positions, although the enzyme is specific to inosine and guanosine. It was also noted in [69] which was about IAG-NH and not 6-oxopurine-specific hydrolase that these tryptophan residues are not preserved. In addition, the alignment also shows that although many purine-specific hydrolases contain semiconservative Phe, Tyr, and Trp residues (e.g., around positions 200, 204, 227), the positions considered in [12] and designated as ring binding of nitrogenous bases, are not conservative. It is possible that Trp, Tyr, or Phe residues are really involved in binding purines, but are located in other positions, or perhaps—even within this small group of hydrolases—each may have its own type of binding and stabilization of the purine ring of the ribonucleoside. The main features of the active site in this case are Asp residues that help bind calcium and ribose, Asn residues that are also involved in ribose binding, and a hydrophobic pocket in which the purine base should be located (possibly the conserved L(T/S)XXA sequence in the region of position 150 is involved in the formation of such a pocket, since it is formed from predominantly hydrophobic amino acid residues and does not have a strict described function in catalysis).

The pyrimidine-specific hydrolases RihA and RihB are found predominantly in bacteria, although these types of ribonucleoside hydrolases are also found in some other organisms, such as plants and protozoa. Appendix A shows the alignment of the amino acid sequences of some of these hydrolases. As in the case of purine hydrolases, these hydrolases have a highly conserved DXDXGXDD sequence at the N-terminus, which is necessary for binding the calcium ion and stabilizing ribose, which is part of the nucleosides [50]. Despite the fact that not only bacterial RihA and RihB were chosen for alignment but also enzymes from plants and yeast, it can be seen that the amino acid sequences of all the presented enzymes contain a large number of highly conserved and semiconserved regions. This may be due to the similarity of the substrates—uridine and cytidine—as well as the fact that there are only two of them. It can be assumed that catalysis proceeds in almost the same way regardless of the source of this type of enzyme. Presumably, highly conserved Asp and Asn residues are required for both calcium binding and ribose binding and stabilization of the transition state during catalysis. Hydrophobic amino acid residues, which are both conservative and semiconservative in these enzymes, are necessary for the formation of a hydrophobic pocket in the active site of the enzyme, among which the following residues are present: Leu, Ile, Val, Ala. In [50], the authors indicate that the binding of the pyrimidine ring in RihB (and accordingly, the same can be assumed about RihA due to the high similarity of these types of hydrolases) involves the Asn(Val/Ile)His sequence in the region of position 140, Phe in 238 position, Phe/Tyr in 311 position, His in 315 position, Tyr/Trp in 320 and His in 329 position. These residues are indeed conserved or semiconserved in RihA and RihB regardless of the source of the enzyme, and may aid in the binding of the nitrogenous base, which is the most hydrophobic part of the nucleoside.

Nonspecific RihC hydrolases have the widest distribution between all types of Rih hydrolases. These hydrolases have been found in a wide variety of organisms, ranging from bacteria to metazoan. Appendix A shows an alignment of the amino acid sequences of a wide range of RihC hydrolases as a basis for studying what unites this type of hydrolase with each other and what is special only for certain organisms. For this type of hydrolase, as in the previous two cases, there is a highly conserved region DXDXXXDD in the region of the N-terminus (for some fungal hydrolases, this region is represented as DXDXXXXXDD), which is necessary for binding the calcium ion and the 2′-OH group of the ribose ring. In general, these hydrolases also contain the Asp and Asn amino acid residues that aid in ribose and calcium binding along the amino acid sequences and hydrophobic regions formed by conserved and semiconservative amino acid residues. It is more difficult to identify key residues in RihC that help to stabilize nitrogenous bases [62], suggesting that His and Tyr residues at positions 404 and 405 and His at position 428 are key to the stabilization of purine bases in RihC from *E. coli*. At the same time, Tyr400, Tyr405, and His428 were proposed in [70] for RihC as key residues binding the purine ring. The same catalytic triad was proposed for the mechanism of catalysis of pyrimidine nucleosides (to stabilize pyrimidine bases). However, paying attention to the alignment, one can clearly see that these residues are not conservative. For some RihC hydrolases, the second tyrosine residue from this triad is indeed retained, and the histidine residue is highly conserved for all RihC except fungal ones. It should be noted that the first residue of the triad does not have a strictly defined position in the amino acid sequence of hydrolases of this type, and in principle, it may not be a strictly conserved His or Tyr residue. However, it has been shown [9,15,16,62,70], that these amino acid residues are required for catalysis. Possibly, the low conservatism in their position can be explained by the difference in the structure of these hydrolases depending on the organism. Evolutionarily, these residues might have “migrated” along the amino acid sequence in one direction or another, depending on the organism. Moreover, the substrate specificity of RihC hydrolases, although generally similar (they can hydrolyze both purine and pyrimidine ribonucleosides), still differs depending on the organism, which may be due precisely to the position of this triad and in turn can be explained by the particular organism in which this RihC hydrolase is produced and what needs have been formed in this organism in the course of evolution in relation to nitrogenous bases and nucleosides.

### 3.2. Rih Crystal Structure

To date, a number of crystal structures of the Rih family enzymes of different types have been obtained. The researchers managed to obtain crystals of recombinant ribonucleoside hydrolases from different organisms: the structures of enzymes from bacteria, archaea, protozoa, and even plants and metazoa are known. Table 2 lists the PDB codes corresponding to the hydrolase structures. Only unique structures are presented in the table. The structures of the same enzyme with different inhibitors/ligands are not shown, and the structures of some mutant forms of enzymes for which wild-type enzyme structures are available are not shown.

Examples of structures are shown in Figure 8 (PDB codes for each structure are in parentheses). It can be seen that Rih hydrolases can be a tetramer (Figure 8A,C) or a dimer (Figure 8B), and the structures can be both symmetric and asymmetric.

All currently known three-dimensional structures of Rih hydrolases contain at least one calcium ion per subunit, which is essential for catalysis [8,15,16,34,70,71,72]. At the same time, some structures have two (Figure 8A) or even three and four (Figure 8C) calcium ions in them. These additional calcium ions may help stabilize the structure itself and not play any role in the catalysis. Figure 9 shows fragments of the active site of IAG-NH from *Trypanosoma vivax* in complex with inosine (Figure 9A) and RihC from *Crithidia fasciculata* in complex with pAPIR (para-aminophenyliminoribitol; Figure 9B). For the IAG-NH enzyme in Figure 9A, one can observe the close arrangement of two tryptophan residues, which, as described earlier, are involved in π-stacking and thus stabilize the nitrogenous base of the nucleoside. Figure 9B shows the close proximity of the histidine and tyrosine residues of the RihC enzyme to the nitrogenous base in the nucleoside (in this case pAPIR plays the role of the nucleoside), which indicates their importance in substrate binding [73]. Both enzymes show the presence of conserved Asp, Asn, Glu, and Thr residues involved in the binding of the calcium ion or the nucleoside ribose ring.

The mechanism of catalysis for purine Rih hydrolases generally corresponds to the mechanism of SN1 nucleophilic substitution with the formation of an oxocarbenium ion in the transition state [11,74]. In this transition state, the N-glycosidic bond of the nucleoside is in a nearly broken state with an interatomic distance of 2Å, while the attacking water molecule is at a distance of approximately 3Å. The study of kinetic isotope effects also showed that protonation of the N7 atom of the purine ring of the nucleoside precedes the appearance of the transition state and leads to destabilization of the N-glycosidic bond. Based on this, three main points in the hydrolysis of purine nucleosides can be distinguished: steric and electrostatic stabilization of the oxocarbenium ion, activation of the nucleophilic water molecule, and activation of the leaving group.

The activation of the water molecule occurs due to its noncovalent binding to the calcium ion in the active center and the pulling of the proton of this water molecule towards itself by the Asp residue. In the noncatalytic solvolysis of nucleosides, it was found that the free OH groups of ribose have a stabilizing effect on the N-glycosidic bond and prevent the formation of a positively charged oxocarbenium ion [75]. For enzymatic catalysis, however, the presence of free OH groups is essential. Presumably, the oxocarbenium transition state is significantly stabilized by the interactions of the metalloenzyme and ribose in two ways. The first is the accumulation of binding energy, which is used to bend the conformation of the ribose ring and bring it into an activated state (steric catalysis). The second is the polarization of some hydroxyl groups with the subsequent appearance of (partial) negative charges on them, which stabilize the resulting positive oxocarbenium ion (electrostatic catalysis).

As noted earlier, the appearance of the transition state is preceded by the protonation of the N7 atom of the purine ring of the nucleoside. During this process, a good leaving group appears, facilitating the destruction of the N-glycosidic bond. The mechanism of such stabilization remains not completely understood, since the Trp amino acid residues are in the immediate environment of the nitrogenous base in the enzyme, but the indole ring of the tryptophan residue cannot be a classical acid that activates the N7 atom, since this ring is not ionized. It was suggested [14] that due to π-stacking, the pK_a_ of the N7 nitrogenous base atom increases, due to which this atom is protonated by water.

In the case of catalysis by pyrimidine or nonspecific hydrolases, the general mechanism is similar to the previous case. Since the calcium ion and conservative aspartic acid residues are present in all enzymes, the activation of the nucleophilic water molecule and the stabilization of the oxocarbenium ion occur in the same way as described above. The activation of the leaving group itself also occurs due to the protonation of the N3 atom of the pyrimidine (analogous to the N7 atom of the purine) nitrogenous base; however, in this case, tyrosine and histidine residues are located near the ring of the nitrogenous base, presumably involved in protonation and the formation of a good leaving group. It is assumed that for RihC hydrolases, during the catalysis of pyrimidine nucleoside cleavage, the protonation of the N3 atom occurs via the histidine residue itself, while in the catalysis of the cleavage of purine nucleosides, the protonation of the N7 atom occurs along the His-Tyr-Tyr chain. At the same time, in the pyrimidine hydrolases RihA and RihB, tyrosine residues are nonconservative; therefore, these hydrolases either cannot catalyze the cleavage of purine nucleosides at all, or they can, but are orders of magnitude worse than the cleavage of pyrimidine nucleosides.

Based on the structure of the active center, namely, depending on the key amino acid residues stabilizing the nitrogenous base of the nucleoside, a classification of Rih nucleoside hydrolases into three groups not based on substrate specificity was proposed (Figure 10). It is customary to include hydrolases in group I that have a histidine residue in their active center (as well as tyrosine residues conjugated with this residue involved in the protonation of the nitrogenous base ring), in group II hydrolases that have a tryptophan residue instead of this histidine residue (and conjugate with it the second tryptophan residue at a distance, forming π-stacking interactions with the purine ring), and in group III hydrolases with a cysteine residue in this position (which usually also has a paired cysteine residue at a distance, also involved in catalysis).

Hydrolases of the first two groups are currently the most studied and most common. In these two groups, there is also a fairly clear division of hydrolases according to their substrate specificity. The third group includes hydrolases with different substrate preferences. Despite the fact that the cysteine residue in the active site is more similar in function to the group I histidine residue and that the protonation of the nitrogenous base occurs in a similar way to the hydrolases of the first group, purine-specific hydrolases have also been found in this group (for example, in [35]).

It should be noted that this classification cannot be considered final, since some Rih hydrolases do not contain any of the described amino acid residues in this catalytically important position, but contain a proline residue. This can be seen in the amino acid alignments described in the previous section for some of the IAG-NH and RihC hydrolases (Appendix A—position 304 on the alignment; Appendix A—position 429 on the alignment). The presence of a proline residue at this position is not a single mutation associated with the evolutionary development of a single enzyme, but is observed for a number of Rih hydrolases from both bacteria and archaea. If for RihC hydrolases from fungi, which have a proline residue in the position binding the nitrogenous ring, a histidine residue is located next to this proline residue, i.e., these changes most likely do not affect catalysis, then for purine-specific hydrolases, there is neither a histidine residue nor a tryptophan residue in the immediate vicinity of this proline residue, which should bind the nitrogenous base in the active center. “Proline” hydrolases have currently not been studied, so the exact mechanism of activation of the leaving group cannot be explained; however, we assume that the mechanism is similar to the activation mechanism in group II: the proline residue increases the pK_a_ of the nitrogenous base and facilitates the protonation of nitrogen by water, which creates a good leaving group. However, to accurately establish the mechanism of catalysis by these hydrolases, experiments and analysis of kinetic data and structure are required.

## 4. Catalytic Properties of Rih Hydrolases

Different types of nucleoside hydrolases have different substrate specificities. The catalytic parameters also differ depending on the type of hydrolase and the organism from which it was isolated. In general, almost all Rih hydrolases studied so far have an optimum catalytic activity at pH 7.0–7.5 and a temperature of 25–37 °C. The exceptions are hydrolases from the thermophilic archaea *Sulfolobus solfataricus* (the temperature optimum of the enzyme was 80 °C), bacteria *Lactobacillus buchneri* LBK78, *Agromyces* sp. MM-1 (the temperature optimum of the enzyme was 50 °C).

Catalytic parameters for all currently studied Rih hydrolases in reactions with natural ribonucleosides are presented in Table 3 and Table 4.

Several patterns can be seen from these tables. Firstly, purine hydrolases are characterized by rather high values of *k_cat_* for their preferred substrates and K_M_ in the range of several (tens in some cases) micromoles. It can be seen that, despite the fact that xanthosine also belongs to purine nucleosides, the rate constants of the enzymatic reaction and the Michaelis constants for the enzymes where the kinetics with this substrate were studied are significantly higher than the constants for the other three purine nucleosides. This may be due to the fact that the additional keto group in position 2 of the purine ring can cause steric hindrance, and since both free oxygen electrons are already occupied in the C=O bond, this keto group is probably poorly stabilized by hydrogen bonds and charged amino acid residues in the active site of the enzyme. At the same time, IAG-NH and IG-NH also react with pyrimidine nucleosides; however, the Michaelis constants in this case are two to three orders of magnitude higher and the catalytic constants two to three orders of magnitude lower, i.e., stabilization of both rings in purine nucleosides due to π-stacking interactions with tryptophan residues in proteins plays an essential role in catalysis. Without such stabilization, as can be seen, the reaction is also possible, but proceeds very poorly.

Secondly, for the studied RihB enzymes related to pyrimidine hydrolases, a similar pattern is observed: with pyrimidine nucleosides, reactions go easily, and K_M_ is already in the order of several hundred micromoles, and with purine nucleosides, reactions either do not happen at all or proceed with catalytic parameters by two to three orders of magnitude worse than those for the reaction with pyrimidine nucleosides. It can be assumed that this is due to the size of the hydrophobic pocket in which the nitrogenous base of the nucleoside is stabilized. It is possible that this pocket is not large enough to accommodate the rings of purine nucleosides, due to which catalysis either does not occur at all or occurs much worse than with pyrimidines (possibly due to conformational changes within the active site and, as a result, the increased distance of catalytically important residues from purine nucleosides when they enter the active site).

Thirdly, for RihC hydrolases, it can be noted that these enzymes are most specific to uridine or inosine, but there is no clear pattern in their preference with respect to other substrates. On average, the K_M_ for RihC is in the order of several hundred micromoles and *k_cat_* varies from a few units to several tens of s^−1^. Bacterial enzymes from *E. coli* and *S. enterica* are characterized by high specificity for uridine. For the enzyme from *E. coli*, it can be seen that all K_M_ are approximately the same; however, *k_cat_* has the highest value in the reaction with uridine. For the enzyme from *S. enterica*, the highest value of *k_cat_* is observed in the reaction with xanthosine; however, the K_M_ for this substrate is almost 10 times greater than the K_M_ for uridine, although their catalytic constants differ by only 1.5 times. Enzymes from *Lactobacillus buchneri* LBK78 and *Agromyces* sp. MM-1 are a special case, because, as shown in the works [58,59], they catalyze reactions with 2′-O-methyluridine. These enzymes are also unique in that they exhibit maximum activity at 70 °C, although their stability is significantly reduced at temperatures above 60 °C, despite the fact that the optimal temperature for the life of organisms from which these enzymes are isolated is 37 °C and 28–30 °C (for different species of *Agromyces*, the temperature optimum of life varies), respectively. At the same time, the activity of these enzymes increases at temperatures from 4 °C to 40 °C and is approximately constant in the range of 40–50 °C, with a sharp increase in activity further up to 70 °C and a sharp drop with even greater heating. This suggests that the function of this hydrolase in these organisms may be in response to stressful conditions, although it is not entirely clear why the highest activity is observed at such high temperatures, at which these organisms should not survive for long.

In addition, in some studies, the kinetic parameters of the discovered nucleoside hydrolases were not studied, but the amount of ribose obtained during the reaction under different conditions was studied instead. For example, in [33] a cell-free extract of the culture of the fungus *Aspergillus phoenicis* was studied for ribonucleosidase activity and activity of purine-specific hydrolase IAG-NH was found, and this enzyme showed the highest activity with inosine under acidic conditions (pH 3.5) with a temperature optimum of 55 °C. At the same time, the organism itself is not thermophilic, but is thermotolerant, which can explain such a high temperature of the optimum functioning of the enzyme. In the same work, it was shown that when extracts of the organism are incubated without the addition of substrates for IAG-NH, the enzyme undergoes thermal inactivation quite quickly; however, under conditions of substrate saturation, the enzyme is stable and best catalyzes the reactions of cleavage of purine nucleosides precisely at this temperature. Similar results were obtained for hydrolases from other fungi, in particular from *Penicillium chrysogenum*, which contains two different hydrolases specific for purine or pyrimidine nucleosides, and from *Fusarium moniliforme*, which contains one nonspecific hydrolase [32]. Hydrolases from these fungi also have a pH optimum in the acidic region (pH optimum for purine hydrolase is 5.0, pH optimum for pyrimidine and nonspecific hydrolases is 6.2), although not in such a strongly acidic environment as the enzyme from [33], and a temperature optimum of 50 °C. At the same time, these fungi are not thermophilic, growth optima are at temperatures of 25–30 °C, and hydrolases, although they show the greatest activity at elevated temperatures, have low thermal stability.

In other works, for example, [26], the activity of the enzyme from *Pisum sativum* (Alaska pea) was studied in general, without determining specific kinetic parameters or the amount of ribose released during the reaction, and found approximately the same activity of the enzyme to different ribonucleosides and even 2′-deoxyribonucleosides. At the same time, the pH optimum for this enzyme was 6.0, which is consistent with the data for plant hydrolases of various types. This enzyme showed the highest activity with adenosine, which suggests its physiological role in plants: regulation of cytokinin concentration through regulation of adenosine concentration. In the work [27], the nucleoside hydrolase activity of the extract from the leaves and roots of *Arabidopsis thaliana* (mouse-ear cress) was studied and two isoenzymes were found, for which the K_M_ was determined in the mixture for inosine, uridine and xanthosine (comprising 200 µM for uridine and inosine and 60 µM for xanthosine). The presence of these hydrolases in the genome of this plant was associated with a high content of uridine in soils where this plant can grow in the wild. However, despite this, both isoenzymes are not physiologically significant for this plant, as was shown in the same work. Plants lacking one or both of the detected ribonucleoside hydrolase genes showed no phenotypic differences from wild-type plants. It is possible that the enzyme is used by the plant in response to stressful conditions, but during normal life it is practically not synthesized and does not play any role in metabolism. Ribonucleoside hydrolases from the muscles of the fish *Ophiodon elongatus* (lingcod) and *Sebastodes* sp. (rockfish) also have interesting properties. Hydrolases from these two organisms have very similar properties, as noted in [37]. These hydrolases are nonspecific and are of the RihC type. At the same time, they show the greatest activity at acidic pH (approximately 5.0) in relation to guanosine and at alkaline pH (about 8.5)–in relation to inosine. At pH 7.0, the activity for all substrates is either at a minimum or almost at a minimum. At the same time, the authors confirm that such unusual properties are not due to different isoforms of the enzyme or even a set of different enzymes, but belong precisely to one type of nucleoside hydrolase. In the case of these hydrolases, as in the case of fungal hydrolases, the maximum activity was achieved at temperatures of 50–55 °C (although the enzyme was inactivated), which again raises the question of the physiological role of enzymes of this type in fish, since the organisms from which the enzymes were obtained do not live at such high temperatures. It is possible that ribonucleoside hydrolases in animals and plants are only residual enzymes inherited from distant ancestors and somehow did not disappear during evolution, but in this case, the question arises of such a great diversity and evolution of these hydrolases: Why did they continue to evolve with organisms instead of remaining approximately the same or disappearing during evolution? Another thing can be assumed: ribonucleoside hydrolases in the normal state are not needed by most multicellular eukaryotes or bacteria, they do not play any key physiological role (the exception is protozoan parasites, where these enzymes, on the contrary, are key in the metabolism of nitrogenous bases); however, these enzymes are involved in the response to stressful conditions for the body.

It is important to analyze two more ribonucleoside hydrolases separately here from the Rih family enzymes. These hydrolases are 1-methyladenosine nucleosidase and 7-methylxanthosine nucleosidase (N-MeNase). This separation is required in our opinion since both of these hydrolases have highly specific reactions with only one substrate: 1-methyladenosine for 1-methyladenosine nucleosidase [39,40] and 7-methylxanthosine for 7-methylxanthosine nucleosidase [41,42,43]. Both enzymes have optimum pH in the range of 7.5–8.5. For 1-methyladenosine nucleosidase, K_M_ was measured [40] and is 665–715 µM, while 1-methylinosine and 1-methylguanosine are also noncompetitive inhibitors for this enzyme [40], with inhibition constants K_I_ of 125 and 140 µM, respectively. The only organisms in which these enzymes have been found to date are starfish for 1-methyladenosine nucleosidase, since it produces 1-methyladenine, which induces oocytes maturation in starfish, and caffeine-producing plants for N-MeNase, since it produces 7-methylxanthine, which turns into caffeine in a few steps.

## 5. Stability of Rih Hydrolases

Typically, enzyme stability to chemical oxidation and temperature stability are determined by the organism that the enzyme is from. Enzymes from mesophilic organisms, as a rule, do not have high thermal stability and are easily oxidized and destroyed in the presence of chemical agents, while enzymes from thermophilic organisms, on the contrary, often have high thermal stability and are less susceptible to oxidation. This is due to the structure of proteins: more thermostable proteins usually have a more rigid structure, i.e., there are covalent cross-links (disulfide bonds of cysteine residues), less mobile elements (a greater number of proline residues), and a greater number of hydrophobic amino acids that provide strong hydrophobic interactions within the protein globule.

As for Rih hydrolases, the patterns of protein stability versus organismal stability persist, although there are some interesting features. As mentioned earlier, *A. phoenicis* IAG-NH hydrolase has the highest catalytic activity with various ribonucleosides at 55 °C [33]. Moreover, under such conditions, the protein is subject to temperature inactivation, and in just 10 min of incubation, loses 50% of its activity in the case of inosine. The organism itself, although not thermophilic, is thermotolerant [29], i.e., the organism’s optimal living conditions are similar to those of mesophilic organisms, but this organism can also withstand higher temperatures. However, in [33], it was shown that when a substrate, for example, inosine, is added to the enzyme, followed by incubation at 55 °C, the thermostability of the enzyme increases; the closed form of the enzyme is more thermostable.

In general, for hydrolases from mesophilic organisms, the optimum catalytic activity is observed at 30–40 °C (with the exception of some hydrolases from thermotolerant organisms, for example, those mentioned above), and when heated above 50 °C, these enzymes lose 50% of their activity in less than an hour [30,32,37,54]. Hydrolases from thermotolerant organisms are usually more thermostable, able to withstand incubation at 50–60 °C without significant loss of activity for an hour or even several hours [33,58,59,63]. Rih hydrolases have also been found in thermophilic organisms. For example, the pyrimidine-specific and purine-specific hydrolases from *Saccharolobus solfataricus* (previously named *Sulfolobus solfataricus*) have an optimum enzyme activity at 100 °C, with enzyme incubation at this temperature for 10 min causing a loss of 10% of activity. Incubation for 10 min at 110 °C results in a loss of 80% of enzymatic activity [50,51,69]. From a structural point of view, this high stability is due to the large number of Leu and Ile residues compared to other hydrolases, as well as the presence of disulfide bonds. These two enzymes are also very stable to the effects of guanidine chloride. The concentration of guanidine chloride, at which half of the enzyme is denatured, is approximately 3.5 M, while denaturation is reversible, and dialysis completely restores enzymatic activity. The enzymes are also stable towards reducing agents such as dithiothreitol (DTT). When exposed to 0.6 M DTT for an hour, the enzymatic activity drops by 40%, which confirms the presence of disulfide bonds which stabilize the structure in the enzyme [51,69].

## 6. Practical Importance of Rih Hydrolases

As mentioned earlier, in most organisms, including mammals, the production of nitrogenous bases occurs either through de novo synthesis or with the participation of phosphorylases during the cleavage of ribonucleosides. Rih hydrolases do not play any significant role in this case. At the same time, trypanosomes and other protozoa are auxotrophs for purines and cannot synthesize them themselves. For these organisms, nucleoside hydrolases play a key role in metabolism and in the production of nitrogenous bases, since nucleoside phosphorylase activity has not been found in these organisms [76]. This difference in metabolism between protozoan parasites and their hosts makes Rih hydrolases a potential target for drug development against these parasites [77].

The first Rih inhibitors were based on the transition state structure of the enzymatic reaction. The best analogues of standard ribonucleosides were iminoribitol derivatives [78]. RihC from *C. fasciculata* was inhibited by phenyl-iminoribitols, and the inhibition constants in the range of a few nanomoles. Iminoribitols associated with purine rings (immucilins) were less effective in inhibiting this enzyme, but inhibited the enzymes IAG-NH and IG-NH quite well with constants in the range of a few nanomoles [79]. The structures of some inhibitors are shown in Figure 11.

However, as it turned out, immucilins are also strong inhibitors of human nucleoside phosphorylase and therefore cannot be used as a therapeutic drug. Although humans have other pathways for metabolizing nucleosides besides the use of nucleoside phosphorylases, these enzymes are key to the body’s production of T cells, and their failure leads to immunodeficiency [80]. In this regard, researchers have been intensively searching for new molecules for the selective inhibition of nucleoside hydrolases. As a result, N-arylmethyl derivatives of iminoribitol were obtained, the use of which gave a selectivity to hydrolases 1000 times higher than the selectivity to phosphorylases [77]. Two such iminoribitol derivatives are shown in Figure 12.

In addition to direct inhibition of protozoan nucleoside hydrolases, there is an approach to use parts of these enzymes in parasite vaccines. For example, Rih from *Leishmania donovanii* is one of the most immunogenic proteins from this organism’s amastigote extracts and is therefore used (as a recombinant protein) in vaccines against murine and canine protozoan diseases [81].

Another approach is based on GDEPT (gene-directed enzyme prodrug activation therapy), i.e., on the enzymatic conversion of a nontoxic prodrug to its active toxic form by an exogenous enzyme expressed in target cells (usually neoplastic cells). Although Rih enzymes are evolutionarily required to catalyze the cleavage of natural ribonucleosides, they can also cleave the N-glycosidic bond in nonnatural counterparts (Figure 13).

The IAG-NH from *Trypanosoma vivax* can cleave 6-methylpurine riboside, releasing a toxic product, 6-methylpurine, which interferes with protein and RNA synthesis. This protein is used in two modern approaches related to the delivery of the prodrug to the right place and the release of the active form of the drug there. The first approach is encapsulation in liposomes with an integrated OmpF bacterial transporter [82]. The second approach is to create a reactor with an amphiphilic three-component copolymer [83]. Both approaches may be promising for cancer therapy.

The RihB enzyme from *E. coli* exhibits catalytic activity towards 5-fluorouridine [72]. The cleavage product of this nucleoside, 5-fluorouracil, is one of the first therapeutic agents used against solid tumors. Cancer cell enzymes convert 5-fluorouracil into the corresponding nucleotides and deoxynucleotides, from which RNA and DNA molecules are subsequently built, leading to a disruption in the assembly of the ribosome. The enzyme thymidylate synthase is inhibited by the 5-fluorouracil derivative fluorodeoxyuridylate (other names: FdUMP, 5-fluoro-2′-deoxyuridine 5′-monophosphate), resulting in thymidine deficiency and subsequent apoptosis. However, 5-fluorouracil is rapidly excreted from the body through the liver, which reduces the number of nucleotides that can be synthesized from it. Currently, 5-fluorouracil derivatives, such as capecitabine, are used in conventional cancer therapy, but such derivatives are not bioavailable. The use of the RihB enzyme for cancer therapy could increase efficacy and bioavailability [72]. If this enzyme is delivered to a tumor and 5-fluorouridine is used as a substrate, then a greater accumulation of 5-fluorouracil can be expected in the tumor than in other tissues, which firstly would reduce the overall toxicity of this substance for the body and secondly would increase the effectiveness of anticancer therapy.

Rih enzymes can also be used in the biotechnological synthesis of important organic molecules as an alternative to conventional chemical synthesis. For example, it has been shown [84] that the use of RihA-expressing *E. coli* cells as a whole-cell biocatalyst is an effective way to obtain large amounts of uracil (30 g/L and more). The use of such catalysts makes it possible to obtain a larger amount of a purer product than organic synthesis without the use of environmentally toxic substances, which is an undoubted advantage.

One more interesting practical use has been proposed for N-methyl nucleosidase. This enzyme, as noted previously, is a key member of a caffeine production pathway from xanthosine [41,42,43]. The researchers in [85] made a transgenic tobacco plant based on *Nicotiana tabacum*, the N-MeNase gene being one of the genes they transplanted and noted that due to caffeine production in this tobacco plant, it started to repel some of the pests, namely, tobacco cutworm *Spodoptera litura*. This method can be used as a novel pesticide method in various crop plants.

## 7. Conclusions

Ribonucleoside hydrolases are an interesting, understudied family of enzymes with potentially useful applications in biotechnological synthesis or in the development and delivery of anticancer drugs. The physiological role of enzymes of this family in some organisms remains not fully understood, so the discovery, cloning, and study of the properties of such hydrolases from different organisms is an important task. In the course of such fundamental research, it is possible to develop new methods for applying these hydrolases in practice or to improve the old ones.

## Figures and Tables

**Figure 1 biomolecules-13-01375-f001:**
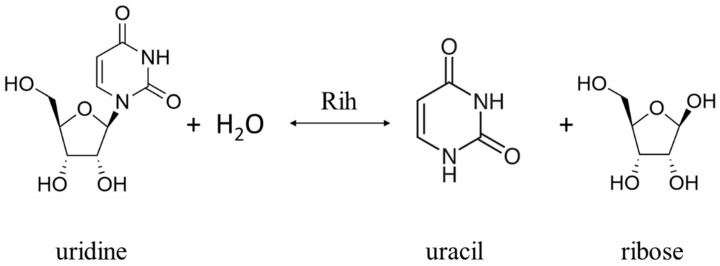
Uridine cleavage reaction by pyrimidine-specific and nonspecific Rih hydrolases.

**Figure 2 biomolecules-13-01375-f002:**
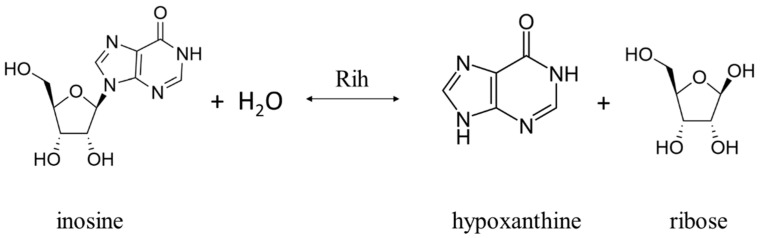
Inosine cleavage reaction by purine-specific and nonspecific Rih hydrolases.

**Figure 3 biomolecules-13-01375-f003:**
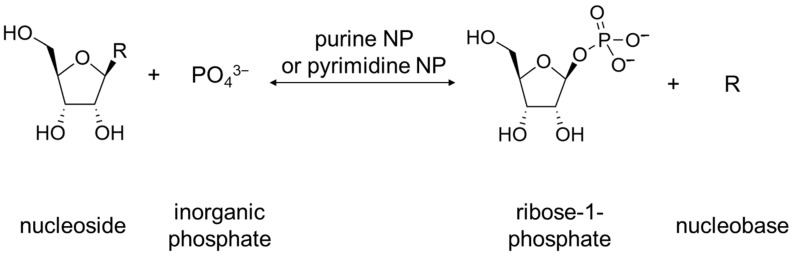
Reaction catalyzed by purine NP and pyrimidine NP. R can be either purine nucleobase or pyrimidine nucleobase.

**Figure 4 biomolecules-13-01375-f004:**
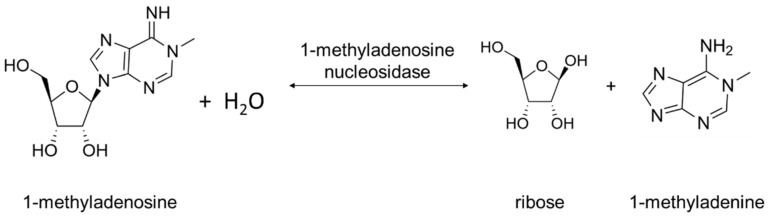
1-Methyladenosine cleavage catalyzed by 1-methyladenosine nucleosidase.

**Figure 5 biomolecules-13-01375-f005:**
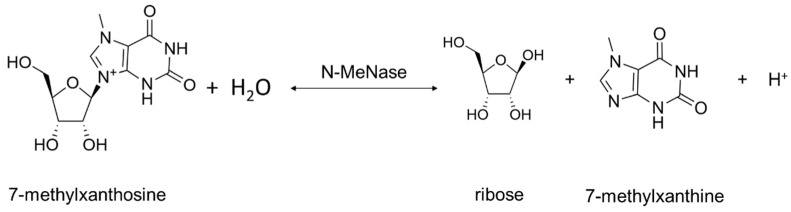
7-methylxanthosine cleavage catalyzed by N-MeNase.

**Figure 6 biomolecules-13-01375-f006:**
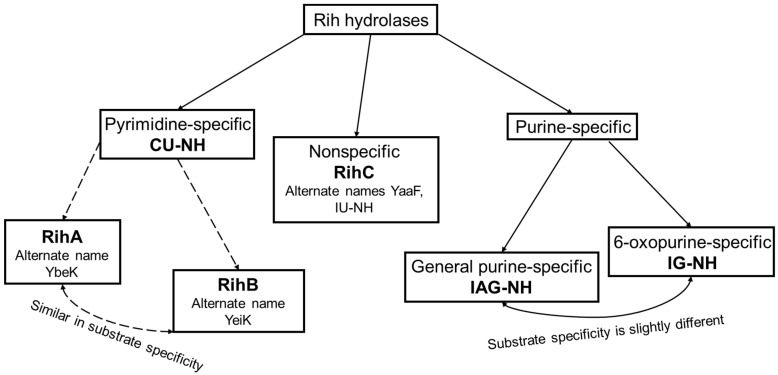
Types of Rih hydrolases.

**Figure 7 biomolecules-13-01375-f007:**
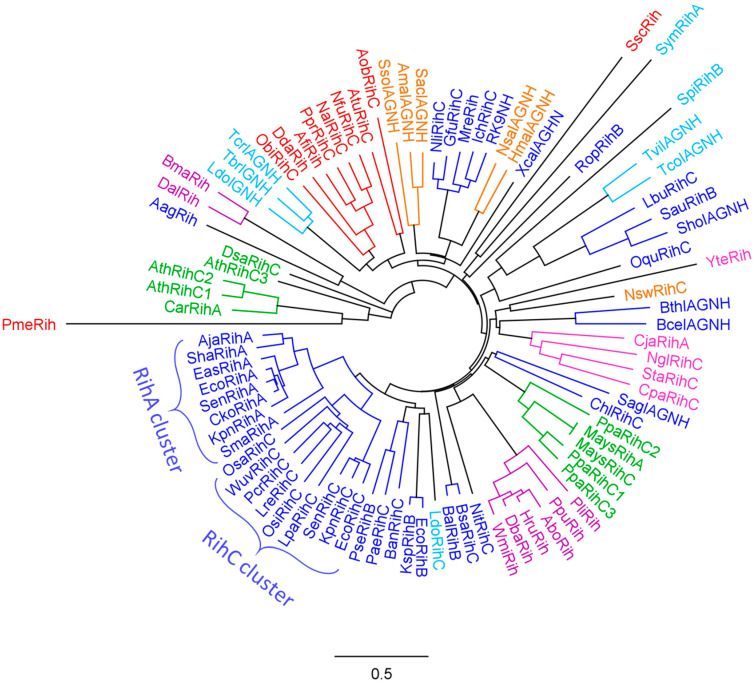
Phylogenetic tree of Rih hydrolases from various organisms. Enzymes from archaea are highlighted in orange, enzymes from bacteria are highlighted in blue, enzymes from fungi are highlighted in purple, enzymes from yeasts are highlighted in pink, enzymes from organisms belonging to metazoa are highlighted in red, from protozoan eukaryotes in light blue, and from plants in green.

**Figure 8 biomolecules-13-01375-f008:**
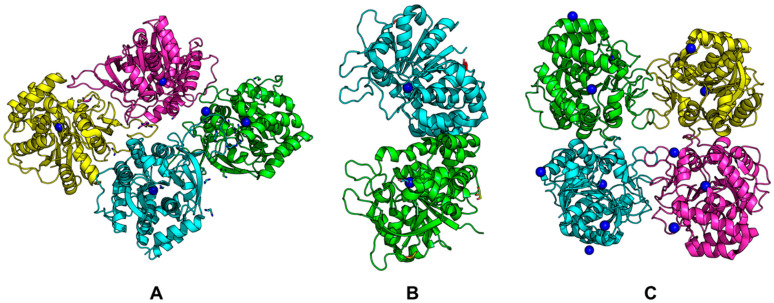
Rih crystal structures. (**A**) RihA from *Zea mays* (6ZK1), (**B**) IAG-NH from *Trypanosoma vivax* (1HOZ), (**C**) RihB from *Gardnerella vaginalis* (6BA0). Calcium ions are shown in blue.

**Figure 9 biomolecules-13-01375-f009:**
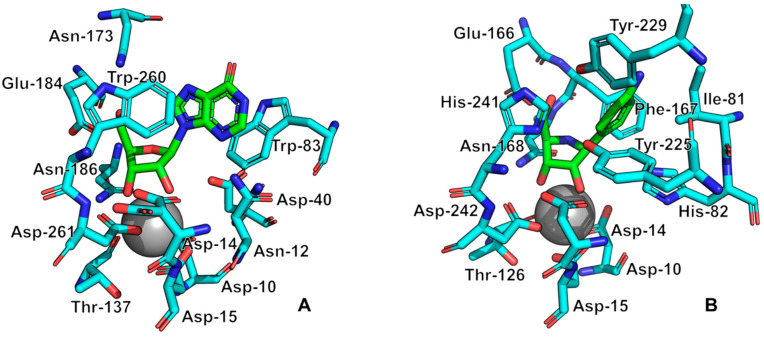
Active site fragments of IAG-NH from *T. vivax* in complex with inosine ((**A**), PDB ID: 1KIC) and RihC from *C. fasciculata* in complex with pAPIR ((**B**), PDB ID: 2MAS). Molecules in the complex with the enzyme are indicated in green, key enzyme amino acid residues are indicated in cyan, calcium ions are indicated by a gray sphere. Key active site residues are numbered for each case.

**Figure 10 biomolecules-13-01375-f010:**
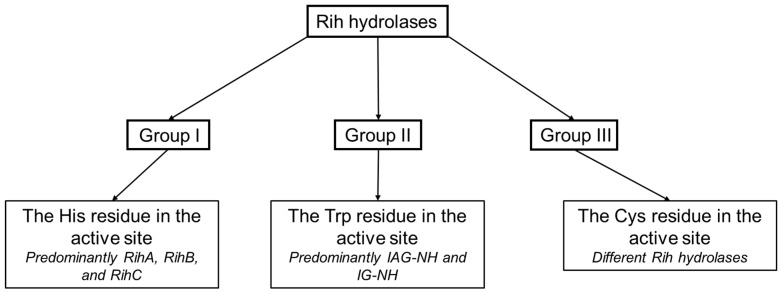
Schematic dividing of Rih hydrolases into three groups depending on the catalytically important residues in the active site.

**Figure 11 biomolecules-13-01375-f011:**
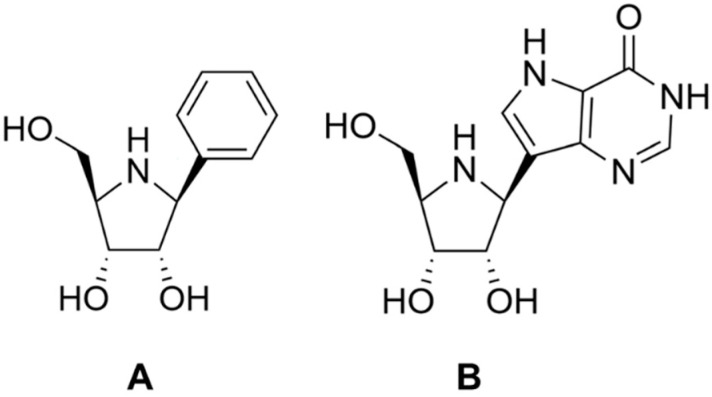
Structure of some early iminoribitol-based Rih inhibitors. (**A**) Phenyl-iminoribitol, (**B**) immucilin H.

**Figure 12 biomolecules-13-01375-f012:**
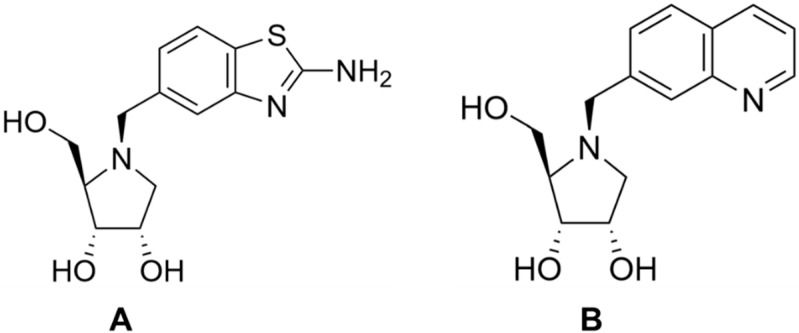
Structure of two iminoribitol derivatives UAMC-00311 (**A**) and UAMC-00115 (**B**) that are 1000 times more selective towards hydrolases than phosphorylases, as shown in [77].

**Figure 13 biomolecules-13-01375-f013:**
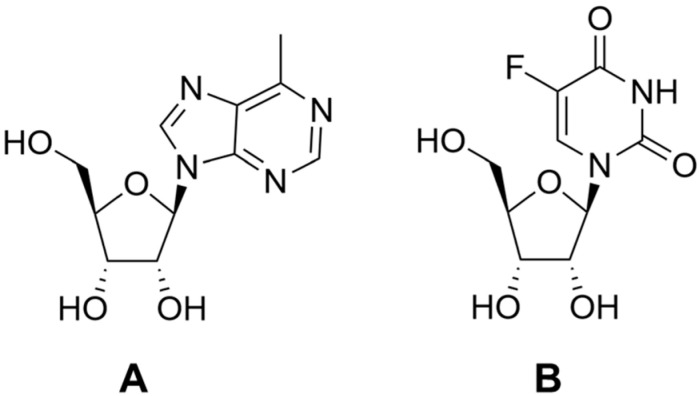
Analogues of natural ribonucleosides: (**A**) 6-methylpurine riboside, (**B**) 5-fluorouridine.

**Table 1 biomolecules-13-01375-t001:** The list of Rih hydrolases used to construct the phylogenetic tree and amino acid alignments. All organisms are color-coded in agreement with the phylogenetic tree. Enzymes from archaea are highlighted in orange, enzymes from bacteria are highlighted in blue, enzymes from fungi are highlighted in purple, enzymes from yeasts are highlighted in pink, enzymes from organisms belonging to metazoa are highlighted in red, from protozoan eukaryotes in light blue, and from plants in green.

Organism Type	Organism	Rih Type	Short Name	NCBI Code
Archaea	* Saccharolobus solfataricus *	IAG-NH	SsoIAGNH	WP_009991400.1
Archaea	* Natrinema salifodinae *	IAG-NH	NsaIAGNH	WP_049989006.1
Archaea	* Acidianus manzaensis *	IAG-NH	AmaIAGNH	ARM75010.1
Archaea	* Natrialba swarupiae *	RihC	NswRihC	TYT60529.1
Archaea	* Haloarcula marismortui *	IAG-NH	HmaIAGNH	QCP93065.1
Archaea	* Sulfolobus acidocaldarius *	IAG-NH	SacIAGNH	AGE70026.1
Bacteria	* Intrasporangium chromatireducens *	RihC	IchRihC	WP_034716916.1
Bacteria	*Nitrospira *sp. OLB3	RihC	NitRihC	KXK05231.1
Bacteria	* Chloroflexi bacterium * OLB14	RihC	ChlRihC	KXK12577.1
Bacteria	* Bacillus safensis *	RihC	BsaRihC	WP_061110295.1
Bacteria	* Pseudomonas aeruginosa *	RihC	PaeRihC	WP_061189914.1
Bacteria	* Brucella anthropi *	RihC	BanRihC	WP_061347523.1
Bacteria	* Limosilactobacillus reuteri *	RihC	LreRihC	MBU5982057.1
Bacteria	* Microbacterium resistens *	*	MreRih	WP_067247535.1
Bacteria	* Nocardiopsis listeri *	RihC	NliRihC	WP_067600215.1
Bacteria	* Periweissella cryptocerci *	RihC	PcrRihC	WP_133363136.1
Bacteria	* Weissella uvarum *	RihC	WuvRihC	WP_205144808.1
Bacteria	* Leuconostoc palmae *	RihC	LpaRihC	WP_220742199.1
Bacteria	* Oceanobacillus salinisoli *	RihC	OsaRihC	WP_156290634.1
Bacteria	* Oenococcus sicerae *	RihC	OsiRihC	WP_128686110.1
Bacteria	* Glycomyces fuscus *	RihC	GfuRihC	WP_097633998.1
Bacteria	* Bacillus thuringiensis *	IAG-NH	BthIAGNH	ABU48602.1
Bacteria	* Escherichia coli *	RihB	EcoRihB	B6I8L3.1
Bacteria	* Citrobacter koseri *	RihA	CkoRihA	A8AJF8.1
Bacteria	* Klebsiella pneumoniae *	RihC	KpnRihC	A6T4G4.1
Bacteria	Metagenomic library	RihC	RK9NH	AYV61512.1
Bacteria	* Salmonella enterica *	RihC	SenRihC	APQ66417.1
Bacteria	* Ochrobactrum quorumnocens *	RihC	OquRihC	WP_095444951.1
Bacteria	* Lentilactobacillus buchneri *	RihC	LbuRihC	AEB72406.1
Bacteria	* Xanthomonas campestris *	IAG-NH	XcaIAGNH	UAU33856.1
Bacteria	* Klebsiella pneumoniae *	RihA	KpnRihA	CDQ54287.1
Bacteria	* Escherichia coli *	RihA	EcoRihA	CAD6019387.1
Bacteria	* Serratia marcescens *	RihA	SmaRihA	OUI66314.1
Bacteria	* Salmonella enterica *	RihA	SenRihA	CAH2854758.1
Bacteria	* Shewanella hafniensis *	RihA	ShaRihA	CAD6364937.1
Bacteria	* Enterobacter asburiae *	RihA	EasRihA	WP_163330465.1
Bacteria	* Aeromonas jandaei *	RihA	AjaRihA	WP_156852166.1
Bacteria	* Klebsiella spallanzanii *	RihB	KspRihB	VUS89946.1
Bacteria	* Rhodococcus opacus *	RihB	RopRihB	CAG7622013.1
Bacteria	* Bacillus altitudinis *	RihB	BalRihB	CAI7725891.1
Bacteria	* Staphylococcus aureus *	RihB	SauRihB	CAG9964796.1
Bacteria	* Pseudomonas * sp. MM227	RihB	PseRihB	CAI3787937.1
Bacteria	* Staphylococcus hominis *	IAG-NH	ShoIAGNH	SUM69402.1
Bacteria	* Streptococcus agalactiae *	IAG-NH	SagIAGNH	CCO74302.1
Bacteria	* Bacillus cereus *	IAG-NH	BceIAGNH	AFQ09851.1
Bacteria	* Anatilimnocola aggregata *	*	AagRih	WP_202921648.1
Bacteria	* Escherichia coli *	RihC	EcoRihC	CAD6022375.1
Fungi	* Daldinia * sp.	*	DalRih	KAI8965458.1
Fungi	* Hypoxylon rubiginosum *	*	HruRih	KAI6091274.1
Fungi	* Annulohypoxylon bovei *	*	AboRih	KAI2472106.1
Fungi	* Poronia punctata *	*	PpuRih	KAI1816947.1
Fungi	* Daldinia bambusicola *	*	DbaRih	KAI1807258.1
Fungi	* Biscogniauxia marginata *	*	BmaRih	KAI1498832.1
Fungi	* Whalleya microplaca *	*	WmiRih	KAI1077167.1
Fungi	* Purpureocillium lilacinum *	*	PliRih	PWI74805.1
Yeast	* Cyberlindnera jadinii *	RihA	CjaRihA	CEP20255.1
Yeast	* Nakaseomyces glabratus *	RihC	NglRihC	KAH7604059.1
Yeast	* Suhomyces tanzawaensis *	RihC	StaRihC	XP_020067128.1
Yeast	* Candida parapsilosis *	RihC	CpaRihC	KAF6065839.1
Yeast	* Yamadazyma tenuis *	*	YteRih	EGV65217.1
Reptiles	* Platysternon megacephalum *	*	PmeRih	TFJ96048.1
Arachnids	* Sarcoptes scabiei *	*	SscRih	UXI17525.1
Actinopterygii	* Anoplopoma fimbria *	*	AfiRih	XP_054471125.1
Actinopterygii	* Dunckerocampus dactyliophorus *	*	DdaRih	XP_054644538.1
Cephalopoda	* Octopus bimaculoides *	RihC	ObiRihC	XP_014774668.1
Insects	* Aethina tumida *	RihC	AtuRihC	XP_019871744.1
Actinopterygii	* Nibea albiflora *	RihC	NalRihC	KAG8004511.1
Actinopterygii	* Nothobranchius furzeri *	RihC	NfuRihC	XP_015807835.1
Actinopterygii	* Pimephales promelas *	RihC	PprRihC	KAG1945547.1
Insects	* Anastrepha obliqua *	RihC	AobRihC	XP_054728927.1
Protozoa	* Trypanosoma brucei brucei *	IG-NH	TbrIGNH	XP_846080.1
Protozoa	* Trypanosoma vivax *	IAG-NH	TviIAGNH	AAG38561.2
Protozoa	* Trypanosoma cruzi *	IAG-NH	TcrIAGNH	EAN96320.1
Protozoa	* Trypanosoma congolense *	IAG-NH	TcoIAGNH	AAG38560.1
Protozoa	* Leishmania donovani *	RihC	LdoRihC	AAG02281.1
Protozoa	* Leishmania donovani *	IG-NH	LdoIGNH	AYU77254.1
Protozoa	* Symbiodinium * sp.	RihA	SymRihA	CAE7031721.1
Protozoa	* Symbiodinium pilosum *	RihB	SpiRihB	CAE7496820.1
Eudicots	* Arabidopsis thaliana *	RihC	AthRihC1	AED92622.1
Eudicots	* Arabidopsis thaliana *	RihC	AthRihC2	AED92625.1
Eudicots	* Arabidopsis thaliana *	RihC	AthRihC3	AED92623.1
Monocots	* Zea mays *	RihA	MaysRihA	NP_001105259.2
Bryopsida	* Physcomitrella patens *	RihC	PpaRihC1	AFZ84928.1
Bryopsida	* Physcomitrella patens *	RihC	PpaRihC2	AFR46616.1
Bryopsida	* Physcomitrella patens *	RihC	PpaRihC3	AFZ84924.1
Monocots	* Zea mays *	RihC	MaysRihC	AFD54993.1
Chlorophyceae	* Dunaliella salina *	RihC	DsaRihC	KAF5829699.1

* The authors did not specify which type of nucleoside hydrolase these sequences corresponded to.

**Table 2 biomolecules-13-01375-t002:** Resolved crystal structures of Rih hydrolases deposited in the PDB database. Enzymes from archaea are highlighted in light orange, enzymes from bacteria are highlighted in blue, enzymes from nematodes belonging to metazoa are highlighted in red, from protozoan eukaryotes in light blue, and from plants in green.

Organism	Hydrolase	PDB Code
* Saccharolobus solfataricus *	IAG-NH	3T8I
* Saccharolobus solfataricus *	RihB	3T8J
* Escherichia coli *	RihA	1YOE
* Escherichia coli * K-12	RihA	3G5I
* Shewanella loihica * PV-4	RihA	4WR2
* Escherichia coli *	RihB	1Q8F
* Escherichia coli * K-12	RihB	3B9X
* Escherichia coli * K-12	RihB	3MKM
* Gardnerella vaginalis * 315-A	RihB	6BA0
* Bacillus anthracis *	RihC	2C40
* Gardnerella vaginalis * 315-A	RihC	6BA1
* Caenorhabditis elegans *	IAG-NH	5MJ7
* Trichomonas vaginalis *	IAG-NH	8DB8
* Trypanosoma brucei *	IAG-NH	4I73
* Trypanosoma vivax *	IAG-NH	1HOZ
* Trypanosoma brucei *	IG-NH	3FZ0
* Crithidia fasciculata *	RihC	1MAS
* Leishmania braziliensis *	RihC	5TSQ
* Leishmania major *	RihC	1EZR
* Zea mays *	RihA	6ZK1
* Physcomitrella patens *	RihC	4KPN
* Zea mays *	RihC	4KPO

**Table 3 biomolecules-13-01375-t003:** Kinetic parameters of Rih hydrolases in reaction with adenosine, guanosine, cytidine, and uridine.

Organism	Rih Type	*k_cat_^uridine^*, s^−1^	K_M_^uridine^, µM	*k_cat_^cytidin^*, s^−1^	K_M_^citidine^, µM	*k_cat_^guanosine^*, s^−1^	K_M_^guanosine^, µM	*k_cat_^adenosine^*, s^−1^	K_M_^adenosine^, µM	Source
*Trypanosoma vivax*	IAG-NH	ND	ND	0.338 ± 0.005	925 ± 39	2.3 ± 0.1	2.3 ± 0.5	2.6 ± 0.2	8 ± 2	[12]
*Trypanosoma vivax*	IAG-NH	0.022 ± 0.002	586 ± 150	0.338 ± 0.005	925 ± 39	1.9	3.82	1.46 ± 0.06	8.54 ± 1.29	[13]
*Trypanosoma cruzi*	IAG-NH	ND	ND	ND	ND	ND	13 ± 2	ND	ND	[2]
*Sulfolobus solfataricus*	IAG-NH	0.060 ± 0.002	3950 ± 500	NA	NA	29.1 ± 0.9	160 ± 16	3.8 ± 0.1	60 ± 8	[52,69]
*Caenorhabditis elegans*	IAG-NH	0.0022 ± 0.0002	2843 ± 813	0.50 ± 0.03	10004 ± 320	0.84 ± 0.09	164 ± 56	0.78 ± 0.05	93 ± 16	[34]
*Trypanosoma brucei brucei*	IG-NH	0.0038 ± 0.0008	1451 ± 600	0.52 ± 0.04	271 ± 49	37.6 ± 0.5	4.5 ± 0.2	0.237 ± 0.005	<1	[4]
*Escherichia coli*	RihB	4.7 ± 0.1	142 ± 8	11.6 ± 1.8	532 ± 101	NA	NA	NA	NA	[72]
*Escherichia coli*	RihB	5.4 ± 0.2	120 ± 10	ND	ND	ND	ND	ND	ND	[70]
*Sulfolobus solfataricus*	RihB	7.1 ± 0.2	310 ± 20	39.4 ± 1.2	970 ± 50	NA	NA	NA	NA	[50]
*Escherichia coli*	RihC	10.85 ± 0.23	408 ± 184	1.12 ± 0.53	682 ± 298	ND	ND	1.15 ± 0.47	416 ± 249	[65]
*Crithidia fasciculata*	RihC	ND	1220 ± 40	ND	4700 ± 500	ND	420 ± 10	ND	460 ± 30	[5]
*Salmonella enterica*	RihC	46 ± 3	1060 ± 100	7.8 ± 1.3	9200 ± 1200	ND	ND	2.06 ± 0.07	160 ± 20	[63]
*Lactobacillus buchneri* LBK78	RihC	3.6 ^a^	320 ± 10 ^a^	ND	ND	ND	ND	ND	ND	[59]
*Agromyces* sp. MM-1	RihC	9.4 ± 0.5 ^a^	3600 ± 400 ^a^	ND	ND	ND	ND	ND	ND	[58]
*Leishmania major*	RihC	32 ± 6	234 ± 112	0.36 ± 0.05	422 ± 175	0.59 ± 0.03	140 ± 23	0.57 ± 0.04	185 ± 46	[8]

^a^ 2′-O-methyluridine was used as a substrate instead of uridine. ND—no data, NA—not applicable, no reaction.

**Table 4 biomolecules-13-01375-t004:** Kinetic parameters of Rih hydrolases in the reaction with inosine and xanthosine.

Organism	Rih Type	*k_cat_^inosine^* (s^−1^)	K_M_^inosine^ (µM)	*k_cat_^xanthosine^* (s^−1^)	K_M_^xanthosine^ (µM)	Source
*Trypanosoma vivax*	IAG-NH	5.19 ± 0.08	5.37 ± 0.42	ND	ND	[13]
*Trypanosoma cruzi*	IAG-NH	ND	18 ± 0.1	ND	ND	[2]
*Sulfolobus solfataricus*	IAG-NH	23.3 ± 0.8	340 ± 20	ND	ND	[69]
*Caenorhabditis elegans*	IAG-NH	0.79 ± 0.04	295 ± 44	0.029 ± 0.002	1038 ± 274	[34]
*Trypanosoma brucei brucei*	IG-NH	28 ± 1	1.9 ± 0.4	6.2 ± 0.3	280 ± 40	[4]
*Escherichia coli*	RihB	NA	NA	ND	ND	[72]
*Escherichia coli*	RihB	0.086 ± 0.003	2340 ± 320	ND	ND	[70]
*Escherichia coli*	RihC	4.31 ± 0.22	422 ± 225	6.30 ± 0.05	454 ± 165	[65]
*Crithidia fasciculata*	RihC	44 ± 3	150 ± 40	ND	ND	[15]
*Salmonella enterica*	RihC	9.0 ± 0.15 ^a^ 8.1 ± 0.14 ^b^	650 ± 60 ^a^ 1280 ± 130 ^b^	62 ± 8 ^a^ 26.3 ± 0.3 ^b^	5900 ± 1000 ^a^ 790 ± 50 ^b^	[63]
*Crithidia fasciculata*	RihC	ND	380 ± 30	ND	ND	[5]
*Leishmania major*	RihC	119 ± 34	445 ± 209	ND	ND	[8]

^a^ Data are given for pH 7.2; ^b^ data are given for pH 6.0. ND—no data, NA—not applicable, no reaction.

## Data Availability

Publicly available datasets were analyzed in this study. This data can be found here: https://www.ncbi.nlm.nih.gov/ and https://www.rcsb.org/. All appropriate ascension codes and PDB IDs can be found in the article.

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
