# Peer review of "Ribonucleoside Hydrolases–Structure, Functions, Physiological Role and Practical Uses"

_biomolecules, 2023, doi:10.3390/biom13091375_

Round 1

Reviewer 1 Report

The manuscript submitted by Shaposhnikov et al. is a summary of current knowledge on ribonucleoside hydrolases. It is an interesting topic and is worth discussing. My first doubt is whether the manuscript is a true review. I would rather consider it as a meta-analysis.

According to my analysis of articles on nucleoside hydrolases in Web of Sciences, the last former review on this topic was published by Wim Versees and Jan Steyaert 20 years ago (Current Opinion in Structural Biology 2003, 13:731–738). This should be clearly stressed in the Introduction. This paper was cited several times further in the text, and these citations should be changed to citations to the original articles (citing a review in a review (?) is not a good practice).

Further comments:
1. In the beginning of the paper, a more detailed comparison of nucleoside hydrolases and phosphorylases (with schemes) should be provided.

2. The authors omitted any mention of N-methyl nucleosidase (EC 3.2.2.25) and 1-methyladenosine nucleosidase (EC 3.2.2.13). They are Rih's as well.

3. All figures need a unification of the style. Now it is a complete mess!

4. The same applies to references. Some are cited as ‘Authors, date’, some as “[number]”. It seems like a rough work, not a manuscript submitted for publication!

5. In the reference list:
      many refs lacks DOI (2, 5, 6, 9, 10-12, 21, 25, 29, 30, 32, 34, 35, 38- 40, 43, 68, 70, 72, maybe more)

      many refs lacks journal names etc. (2, 6, 11, 12, 21, 30, 34, 35, 38, 72, maybe more)

      some refs have strange signs or numbers (4, 11, 70, maybe more)

      most refs cite abbreviated journal names but some cite full names (e.g., 77)

      ref 29 is a journal published by a ‘predatory publisher’ (Academic Journals); citing such journals is not acceptable since they do not have a scientific value

      refs 49 & 60 are Master of Science thesis; scientific value of such works is also doubtful (important results would be published in scientific papers)

      ref 16 requires more precise data (publisher, ISBN) and page numbers

      ref 5: why a part of the title is in capital letters?

      To summarize this point: the authors should carefully check all references, add missing data, and remove some of them.

6. Page 3: The sentence “Nucleoside hydrolase from the Aedes aegypti mosquito, which is a carrier of Dengue fever virus, is injected into the body at the site of a mosquito bite, destroying circulating adenosine, thereby preventing activation of mast cells through purinergic signaling and thus acting as a local anesthetic” requires a reference. The authors should carefully check if all information is supported with references.

7. Page 3: “Currently, hydrolases of this type are called RihA and RihB” – there should be an explanation here on the difference between RihA and RihB. Otherwise, the reader might be confused.

8. Page 3: There should be here a note about nucleosidases EC 3.2.2.25 & EC 3.2.2.13, and the reactions catalyzed by them:

N-Methyl nucleosidase (EC 3.2.2.25): 7-methylxanthosine + H2O <= >7-methylxanthine + D-ribose

1-Methyladenosine nucleosidase (EC 3.2.2.13):  1-methyladenosine + H2O <=> 1-methyladenine + D-ribose

9. Page 2: “According to this, ribonucleoside hydrolases do not play a key role in the metabolism of bacteria, although they are present in their genomes”. Enzymes in genomes?

10. Page 11: “All currently known three-dimensional structures of Rih hydrolases contain one calcium ion per subunit [66]” Is it really one? In Fig. 5C there are 3-4 Ca ions in a subunit. Correction or explanation is required.

11. Page 20: ‘As a result, N-arylmethyl derivatives of iminoribitol were obtained’ - the chemical structures of these compounds should be provided.

All of these notes indicate that the manuscript was in part in a rough form and requires significant corrections to be accepted for publication.

In general, the language is correct, although there are some minor errors. They are not significant.

Author Response

Response to Reviewer 1 Comments

Point 0: According to my analysis of articles on nucleoside hydrolases in Web of Sciences, the last former review on this topic was published by Wim Versees and Jan Steyaert 20 years ago (Current Opinion in Structural Biology 2003, 13:731–738). This should be clearly stressed in the Introduction. This paper was cited several times further in the text, and these citations should be changed to citations to the original articles (citing a review in a review (?) is not a good practice)

Response 0: We added the brief information about the last Rih review in Introduction on page 2 as suggested by the Reviewer. We pointed out the differences in that review (since it only focused on Rih structure and mechanism of catalysis) and our work. We also removed any citations to it throughout the text and changed them with the respective articles used in that review.

Point 1: In the beginning of the paper, a more detailed comparison of nucleoside hydrolases and phosphorylases (with schemes) should be provided

Response 1: We added a reaction catalyzed by nucleoside phosphorylases and information about their involvement in the salvage pathway on page 3

Point 2: The authors omitted any mention of N-methyl nucleosidase (EC 3.2.2.25) and 1-methyladenosine nucleosidase (EC 3.2.2.13). They are Rih's as well.

Response 2: We added information on these nucleoside hydrolases on page 4 as well as two reactions they catalyze, information about kinetics on page 19, and also a practical use of N-MeNase on page 22

Point 3: All figures need a unification of the style. Now it is a complete mess!

Response 3: All new and old figures have been unified in style where applicable. All chemical reactions and organic molecules were redrawn in a more cohesive way. Schematics on Rih division by substrate specificity or catalitically important amino acid residue were made black-and-white to make them more cohesive with the rest of the figures

Points 4 and 5: The same applies to references. Some are cited as ‘Authors, date’, some as “[number]”. It seems like a rough work, not a manuscript submitted for publication!

Response 4 and 5: We apologise for errors in the References. Our reference manager didn’t upload all references correctly the first time and we didn’t double check that. We thank the Reviewer for pointing that out. We made sure that all references have DOI where applicable (except for references on web sites, one reference on the book and the reference 19 which we couldn’t find DOI for), have correct names and journals now. We added information on publisher and ISBN for reference 16 (now reference 17) and appropritate page numbers in text when discussing de novo and salvage pathways (page 3 of this work). We thank the Reviewer for pointing out the predatory publisher work and as such we removed any references to this work from our text. We also removed references to Master of Science theses as suggested by the Reviewer since we couldn’t find the related published scientific articles on the same topic and those theses were not critical for our review.

Point 6: Page 3: The sentence “Nucleoside hydrolase from the Aedes aegypti mosquito, which is a carrier of Dengue fever virus, is injected into the body at the site of a mosquito bite, destroying circulating adenosine, thereby preventing activation of mast cells through purinergic signaling and thus acting as a local anesthetic” requires a reference. The authors should carefully check if all information is supported with references

Response 6: We added the appropriate reference on (now) page 4

Point 7: Page 3: “Currently, hydrolases of this type are called RihA and RihB” – there should be an explanation here on the difference between RihA and RihB. Otherwise, the reader might be confused.

Response 7: We added the appropriate information on this on (now) pages 4 and 5

Point 8: Page 3: There should be here a note about nucleosidases EC 3.2.2.25 & EC 3.2.2.13, and the reactions catalyzed by them

Response 8: We added the appropriate information with reactions on (now) page 4

Point 9: Page 2: “According to this, ribonucleoside hydrolases do not play a key role in the metabolism of bacteria, although they are present in their genomes”. Enzymes in genomes?

Response 9: We made the correction on (now) page 3

Point 10: Page 11: “All currently known three-dimensional structures of Rih hydrolases contain one calcium ion per subunit [66]” Is it really one? In Fig. 5C there are 3-4 Ca ions in a subunit. Correction or explanation is required.

Response 10: We made the correction on (now) page 12

Point 11: Page 20: ‘As a result, N-arylmethyl derivatives of iminoribitol were obtained’ - the chemical structures of these compounds should be provided.

Response 11: We added the appropriate structures on page 21

Reviewer 2 Report

Review Report 1 on “Ribonucleoside hydrolases – structure, functions, physiological role and practical uses”

Recommendation

Minor Revision

Comments to Author:

Manuscript ID: biomolecules-2584521

Type: Review

Title: Ribonucleoside hydrolases – structure, functions, physiological role and practical uses.

Overview and general recommendation:

In this review, the authors collected useful information on the enzyme class of ribonucleoside hydrolases.

The paper is a valuable support for researchers working with these enzymes since it covers different aspects in an effective and exhaustive way and, to the best of my knowledge, there are no specific reviews on ribonucleoside hydrolases in literature.

This comprehensive review is well-written, a great number of studies were addressed and the main features of ribonucleoside hydrolases were clearly presented. Appropriate comments and assumptions were also made based on literature data.

Therefore, I believe this work is suitable for publication after addressing a few minor points.

Comments:

- On page 3, I suggest to add the extended form of the acronym CU-NH in the text, as the authors did for the other acronyms (IAG-NH, IG-NH, IU-NH).

- In my opinion, details on colour code should be inserted in Table 1 and 2 captions instead of the main text.

- I could not find captions for Figures S1, S2 and S3. Please, insert them in the Supplementary files. In addition, I propose to move details such as those presented on page 9 (Alignments were performed using the MUSCLE algorithm (https://www.ebi.ac.uk/tools/msa/muscle/), visualization was carried out in the Geneious Prime program (http://www.geneious.com/). Blosum62 was used as a coincidence matrix. The color on each of the alignments marks amino acid residues that have a certain percentage of coincidence: green - ≥90% coincidence, yellow - 80-90%, orange - 60-80%) in the captions.

Author Response

Response to Reviewer 2 Comments

Point 1: On page 3, I suggest to add the extended form of the acronym CU-NH in the text, as the authors did for the other acronyms (IAG-NH, IG-NH, IU-NH)

Response 1: We added an explanation to the abbreviation CU-NH on (now) page 4

Point 2: In my opinion, details on colour code should be inserted in Table 1 and 2 captions instead of the main text

Response 2: We moved details on color code to Table 1 and Table 2 captions and removed them from the main text as suggested by the Reviewer

Point 3: I could not find captions for Figures S1, S2 and S3. Please, insert them in the Supplementary files. In addition, I propose to move details such as those presented on page 9 (Alignments were performed using the MUSCLE algorithm (https://www.ebi.ac.uk/tools/msa/muscle/), visualization was carried out in the Geneious Prime program (http://www.geneious.com/). Blosum62 was used as a coincidence matrix. The color on each of the alignments marks amino acid residues that have a certain percentage of coincidence: green - ≥90% coincidence, yellow - 80-90%, orange - 60-80%) in the captions

Response 3: We made three .txt files for each of the supplementary Figures that contain captions for each of the Figure. These .txt files are name “Figure S1 caption”, “Figure S2 caption”, and “Figure S3 caption”. We also added the details on alignments in those captions as suggested by the Reviewer and removed them from main text

Round 2

Reviewer 1 Report

The manuscript is now suitable for publication with the exception of the structures. All riboses and their derivatives have a wrong stereochemistry – these are L-riboses instead of D-riboses. So all "up-bonds" (solid wedges) should be changed to "down-bonds" (dashed wedges), and vice versa,  all "down-bonds" (dashed wedges) should be changed to "up-bonds" (solid wedges).

Author Response

Thank you for your comment. We have added the correct ribose structures.